# How is Respect and Social Inclusion Conceptualised by Older Adults in an Aspiring Age-Friendly City? A Photovoice Study in the North-West of England

**DOI:** 10.3390/ijerph17249246

**Published:** 2020-12-10

**Authors:** Sara Ronzi, Lois Orton, Stefanie Buckner, Nigel Bruce, Daniel Pope

**Affiliations:** 1Department of Health Services Research and Policy, Faculty of Public Health and Policy, London School of Hygiene and Tropical Medicine, London WC1H 9SH, UK; 2Department of Public Health, Policy and Systems, University of Liverpool, Liverpool L69 3DT, UK; l.orton@sheffield.ac.uk (L.O.); nigelbruce16@outlook.com (N.B.); danpope@liverpool.ac.uk (D.P.); 3Department of Sociological Studies, University of Sheffield, Sheffield S10 2TU, UK; 4Cambridge Public Health, University of Cambridge, Cambridge CB2 0SR, UK; sb959@medschl.cam.ac.uk

**Keywords:** Age-Friendly Cities, photovoice, qualitative research, healthy ageing, active ageing, older people, social inclusion, participation, UK

## Abstract

The World Health Organisation (WHO) Global Age-Friendly Cities (AFC) Guide classifies key characteristics of an AFC according to eight domains. Whilst much age-friendly practice and research have focused on domains of the physical environment, those related to the social environment have received less attention. Using a Photovoice methodology within a Community-Based Participatory Research approach, our study draws on photographs, interviews and focus groups among 26 older Liverpool residents (60+ years) to explore how the city promotes respect and social inclusion (a core WHO AFC domain). Being involved in this photovoice study allowed older adults across four contrasting neighbourhoods to communicate their perspectives directly to Liverpool’s policymakers, service providers and third sector organisations. This paper provides novel insights into how: (i) respect and social inclusion are shaped by aspects of both physical and social environment, and (ii) the accessibility, affordability and sociability of physical spaces and wider social processes (e.g., neighbourhood fragmentation) contributed to or hindered participants’ health, wellbeing, intergenerational relationships and feelings of inclusion and connection. Our findings suggest that respect and social inclusion are relevant across all eight domains of the WHO AFC Guide. It is core to an AFC and should be reflected in both city-based policies and evaluations.

## 1. Introduction

### 1.1. Age-Friendly Cities Initiatives

Population ageing and urbanisation present unique public health challenges requiring urgent action [1]. Involving older adults in creating social and physical environments that better support healthy ageing is important for public health policy [2]. The past fifteen years have seen a proliferation of efforts across the world to create environments that are ‘age-friendly’ (see References [2,3,4,5]). Many of these endeavours have concentrated on urban environments, although a parallel focus on ageing in rural settings has also emerged [6,7,8,9]. In this study, an Age-Friendly City (AFC) or environment is defined as one that “encourages active ageing by optimising opportunities for health, participation and security in order to enhance quality of life as people age” [10] (p. 1), and where “policies, services, settings and structures support and enable people to age actively” (ibid., p. 5). In line with the World Health Organisation (WHO) [10] definition, we use the term ‘older adults’ to refer to people aged 60+ years.

The WHO has played a leading role in the development of AFCs, producing a range of products from planning and implementation guidance and outcomes indicators (e.g., References [10,11,12,13]). Launched in 2006, the AFC Initiative [10] classified the key characteristics of an AFC according to eight core domains (Figure 1a).

The WHO AFC guide is based on appraisal of the evidence and global empirical research involving older people and city stakeholders (Vancouver Protocol) [15]. Over time, the AFC domains have evolved (Figure 1b), highlighting three themes that cut across the domains of age-friendly environments—the physical and social environment, and municipal services. The WHO domains aim to represent the multifaceted nature of age-friendliness in complex real-world environments. Although presented as distinct elements, the domains overlap and are interlinked [7]. Treating them as distinct elements, however, provides a useful analytical framework for exploring important elements within each domain and when using the AFC Guide for purposes of evaluation benchmark of AFCs [13].

The AFC Guide has become one of the most commonly employed instruments to evaluate the age-friendliness of cities, and it continues to guide initiatives globally [14,16,17,18,19]. Since the publication of the Guide, other age-friendly frameworks and definitions have been developed that reflect the varying approaches to, and organisations participating in, the development of age-friendly environments [16,18,20,21,22]. Some have focused more on the physical structure of an AFC (e.g., housing, accessible pavements and outdoor spaces), while others have focused on the social aspects of an AFC (e.g., social participation) [23].

### 1.2. Respect and Social Inclusion

Whilst much age-friendly practice and research have focused on the domains that relate to the physical environment (e.g., housing, outdoor spaces and transportation) [24,25,26,27,28], those related to the social environment (e.g., respect and social inclusion) have received less attention [20,29,30,31]. This leaves an important gap in knowledge to inform policy and practice on healthy ageing.

Among the studies that have examined social aspects of the urban environment, Woolrych et al. [32] used semi-structured interviews in three cities and nine neighbourhoods in the United Kingdom, which shed light into how older adults constructed and negotiated elements of social participation within their everyday urban environments. In another study [33], qualitative photoproduction conducted by researchers of two municipalities in the Netherlands explored aspects of the built environment that reinforced age-stereotypes and ageism explicitly or implicitly—using the eight WHO AFC domains as an analytical framework. However, an important limitation of the study is that older people were not directly involved in the photo-production process, with photographs taken and disseminated (to health professionals and older adults) by the research team. In evaluating age-friendly programmes in the USA, Scharlach and Lehning [20] highlighted the potential for initiatives to harness communities in promoting the respect and social inclusion of older adults. Buffel et al. [34], through policy document review and secondary qualitative data from interviews with stakeholders, considered the extent to which AFC initiatives in Brussels, Dublin and Manchester were reducing social exclusion in older people. Findings showed that in each city, AFC initiatives had been designed to reduce single or multiple areas of social exclusion (e.g., promoting participation, tackling social isolation and reducing neighbourhood exclusion). Although with some overlaps, their study focused on aspects of social exclusion, rather than on the WHO AFC domain of respect and social inclusion.

With regards to the importance of making older adults feeling valued and respected, persistent disrespectful attitudes, and misconceptions about ageing, have been identified as significant barriers to the development of effective public health policies on healthy ageing [35,36]. These barriers have consequences for the way ageing is perceived and can negatively impact on the health and wellbeing of older adults [37]. According to Swift and Steeden [38], societal attitudes towards older adults can take the form of ‘benign indifferences’, with ageism tending to manifest indirectly, for instance, as lack of respect. Ageism can be defined as “a combination of how we think about age (stereotypes), how we feel about age (prejudice) and how we behave in relation to age (discrimination)” [39] (p. 2). Research on what makes people feel valued and respected as they age can therefore provide insights that are of value in combating ageism, a strategic objective in the *WHO’s Global Strategy and Action Plan on Ageing and Health* [39]. This Action Plan urges a move away from problematising ageing towards a positive respectful and inclusive perspective of older adults that highlights their contributions. It advocates an approach to healthy ageing based on a salutogenic model that focuses on aspects that make people healthy [30,40]. 

Building on existing literature, our study focused on examining the WHO AFC social domain of respect and social inclusion. We define respect and social inclusion as ‘enhancing the opportunities for people of all ages to (i) cultivate social relationships, (ii) have access to resources and support, (iii) feel valued and respected and (iv) feel part of their community’ [20]. Urban environments have an important role in fostering respect and social inclusion by being “socially inclusive of all people—regardless of age, gender, social position, health or disability—[ensuring they] are respected and have opportunities to participate and contribute” [14] (p. 69). A specific commitment must be to create interventions that reach those most in need, including those at risk of social exclusion and ill health. Supporting a community to foster respect and social inclusion can have important public health benefits. In a systematic review of qualitative and quantitative studies conducted in high-income countries, Ronzi et al. [41] described positive associations between interventions promoting respect and social inclusion and a variety of health outcomes among older adults aged 60 and over, including a positive impact on wellbeing, subjective health, quality of life and physical and mental health.

### 1.3. Study Aim

This paper is based on doctoral research by S.R. conducted in Liverpool, UK (2013–2016). It draws on data collected from older residents across four contrasting neighbourhoods in the city. We employed a Photovoice methodology within a Community-Based Participatory Research approach to: (i) explore the extent to which respect and social inclusion were promoted as the city sought to become more age-friendly/an AFC, and (ii) actively involve older adults in the research process and to allow them to directly communicate issues with stakeholders involved in AFC policy.

Previous photovoice studies have explored positive and negative factors for promoting age-friendliness in both rural and urban environments [33,42,43]. These factors have included aspects related to physical activity [44], ability to walk outside [45], access to green spaces [26], perceptions of ageing-in-place [46], places and health [47] and social participation [48]. However, none have focused primarily on respect and social inclusion, as a crucial domain of the WHO AFC Guide.

## 2. Materials and Methods

### 2.1. Study Setting

The study took place in Liverpool, North West of England (UK). Over the years, as many cities in the UK, Liverpool has seen a notable increase in its population of older adults. According to latest data (June 2020), the number of people aged 65+ years is approximatively 73,514 [49], with a projected increase to 96,400 by 2035 [50]. This has important implications not only for promoting healthy ageing, but also for the increased urbanisation in Liverpool.

In an ageing population, the years of life gained do not necessarily equate to healthy-life years. Liverpool’s residents have worse health than the rest of England, with the city continuing to be ranked among the most deprived local authorities in the UK, according to The English Indices of Deprivation 2019 [51]. Inequalities in health are also clearly seen at a local level, with as much as a 12-year gap in life expectancy at birth between Liverpool’s least and most deprived wards [51]. Addressing these health inequalities should therefore be central to AFC policies designed to enable positive ageing experiences.

### 2.2. Participants

Four groups of older adults aged 60+ years (*n* = 26) were recruited from four geographical areas with contrasting socioeconomic conditions in Liverpool, to explore potentially different experiences of respect and social inclusion in the city. Criteria for recruitment are described in Table 1.

We included anyone who was able to communicate fluently in English. This included those older people of non-British heritage able to speak English fluently. We required fluency in English (criterion 3) as, due to budget constraints, we were not able to have an interpreter. We recruited people who had lived in the UK for at least 10 years, for participants to provide insightful information about respect and social inclusion based on their familiarity with the City. Potential participants who did not meet these criteria were excluded.

Participants were recruited through connections that S.R. developed with gatekeepers working in local grassroots organisations in each of the four geographical areas. Grassroots organisations are typically small and work directly with the community in the surrounding area. Such organisations played an essential role in helping S.R. to recruit a mix of more and less socially included older people, by reaching those who were less actively involved in the community. One organisation, in particular, targeted people of all ages who were less embedded in the community for several reasons (e.g., mental and/or physical disability and/or socio-economic difficulties) and provided transportation to those not able to use public transportation.

Gatekeepers were provided with details of the study inclusion criteria (Table 1) and personally introduced S.R. to prospective participants. They also advised on other aspects of the data collection process (e.g., preferred times and days to conduct focus groups). S.R. spent between four to eight months building a trusting relationship with prospective participants through informal visits during weekly activities at each centre where recruitment of participants was planned. Such relationship building was also important in understanding the context in which older adults were living and to explore and understand the dynamics of the participant group. S.R. explained the purpose of the study to the community centre users and assessed their eligibility using a set of open questions that reflected the inclusion criteria. Those meeting all inclusion criteria were invited to take part in the initial focus group and provided with an information sheet and a consent form. Participants included four people with limited mobility (e.g., use of the stroller and walking sticks). All participants were informed that assistance with taking photographs was available from S.R. if required. Three participants were accompanied to places they wanted to photograph, and they were assisted with setting up the camera.

### 2.3. Photovoice

One of the key mechanisms necessary to build AFCs and create inclusive environments is the active involvement of older adults in identifying priorities for action and in decision-making processes [2]. Doing so not only helps older adults to express their views, but it also ensures that developments for AFCs respond to their needs [30,52].

An interpretivist phenomenological approach guided our study [53], which focused on identifying meanings and perceptions that older people had of respect and social inclusion in the urban context. We employed a Photovoice methodology within a Community-Based Participatory Research (CBPR) approach, focused on participation, action and collaborative investigation [54,55]. Photovoice, a methodology developed by Wang and Burris [56], involves participants photographing aspects of their lives and communities that are important to them, with group discussions being held to facilitate critical reflection about community strengths and issues [57,58]. Photovoice was used to provide a unique perspective on the issue of respect and social inclusion in the urban context whilst creatively involving participants in the research process.

A modification of Wang and Burris’ [56] original methodology was adopted, adding individual semi-structured interviews (SSIs) to the process in addition to focus group discussions (FGDs) that are used for photovoice. The addition of SSIs has previously been reported by, for example, Novek et al. [59]. The SSIs helped to (i) build a relationship with each participant and (ii) examine the individual views of participants in relation to the photographs they had taken in greater depth than could be achieved through FGDs, the latter focusing on stimulating discussions around the photos. Each photovoice project (Phase 1–4, Table 2) lasted about one month and a half and was repeated separately for each group. At the end of the study, each participant received a supermarket voucher (valued £20) as a thank you for their time.

### 2.4. Data Analysis

The data presented in this paper come from the photographs (*n* = 127), FGDs (*n* = 8) and SSIs (*n* = 21 individual interviews; *n* = 2 interviews were conducted in pairs, with a total N = 25 participants being interviewed). SSIs and FGDs were transcribed and anonymised, and imported into NVivo 11 software [66] for analysis.

Techniques from thematic analysis [67] were used and focused on identifying meanings and perceptions that older people had of respect and social inclusion in the urban context. S.R. (i) reviewed the transcripts one-by-one, applying provisional thematic codes, and (ii) incorporated these into emerging sub-themes. L.O. double coded 20% of the transcripts for accuracy. Any discrepancy was resolved by discussing with a third reviewer (D.P./N.B.). S.R. iteratively refined sub-themes and themes as the analysis progressed. Over the course of focused analysis sessions, data were discussed, and themes adapted and agreed with the other members of the research team. Whilst data from the FGD transcripts offered a group view on respect and social inclusion (Phase 1 and Phase 4, Table 2), data from the SSIs provided an individualised and often more intimate view, offering in-depth information about the context, meaning and importance associated with the chosen photographs (Phase 3, Table 2).

According to Wang and Burris [56], the meaning of the images resides in the ways that participants interpret those images. Therefore, photographs and their associated meanings were considered within the context of the corresponding transcripts and not analysed separately. The meanings of each photograph was captured during the SSIs and FGD2, which were guided by the visual images (Phase 2 and Phase 4, Table 2) [25,68]. Further details about the data analysis and the link between transcripts and photos are included in our companion paper [64].

## 3. Results

The demographic composition of the study sample is shown in Table 3.

### 3.1. Overview

Older adults highlighted both contrasting and complimentary aspects of the urban context that fostered or hindered perceived respect and social inclusion, and ways in which some of the barriers identified could be reduced. Overall, the combination of accessibility, affordability and sociability of physical spaces contributed to older adults’ mental wellbeing, feelings of inclusion, sense of independence and connection. Participants also identified aspects of the physical (e.g., litter on the streets) and the social environment (e.g., a lack of respect for the older generation and people living in more disadvantaged areas) that contributed to feelings of vulnerability and a sense of exclusion. Wider social processes (e.g., family and neighbourhood fragmentation) were reported to lead to difficulty in cultivating intergenerational relationships.

To facilitate the presentation of the results, we have structured the different sub-themes that emerged from the analysis around the three key dimensions of an AFC, as described by the revised WHO AFC framework (Figure 1b): physical environment, social environment and services.

### 3.2. Physical Environment

#### 3.2.1. Green and Blue Spaces

Green and blue spaces were the most photographed and talked about places by participants across the four groups. They offered (free) opportunities to do physical activity and provided a space for multi-generational interactions (Figure 2).
“We have so many good open spaces with trees and water around the city… and they’re really good for your mental health and wellbeing… and feeling included, as you mix with everybody in a park! It gives you interest, it gets you out, and it gets you in the open area. Sefton Park is a focal point for life in the city. It is a real bonus that we have. We’re lucky!”(P1, M, 65, Group 1) (Figure 2)

Paved, flat and accessible walking paths and parks were believed to be particularly important not only for older people, but for everyone with functional limitations or disabilities (Figure 3).
‘This is the Promenade, it’s a great facility. […] it’s all on the flat for everyone to go for a walk. There are no hills involved… and you’ve got the nice aspect of the river... It’s a lovely facility to have and it’s used by lots of different age groups.’(P17, F, 64, Group 3) (Figure 3)

As shown by these examples, an important aspect that made these green spaces ‘inclusive’ was that all age groups could access and use them. The quote below, however, highlights that the geographical location of many green spaces in Liverpool is in the most affluent parts (e.g., the South of Liverpool) and may not be so easily accessible to everyone.
*‘People of all ages can access it, there’s parking, so you can walk down and on a nice summer’s day people of all ages are out, […]…it’s a good place to go and have a good walk and take the fresh air.…we’re lucky in the South end of Liverpool that we have something like this’*.(P9, M, 71, Group 2) (Figure 4)

Alongside parks, allotments and leisure facilities encouraged physical activity, by providing a space for people to walk, exercise and volunteer. They also had an indirect impact on wellbeing due to their sociability—providing a space for people to come together and cultivate hobbies (Figure 4).
‘I have one of those plots in the community garden. It’s an occupation that I enjoy. […] It’s engaging with others, there’re all sort of people involved in this. […] from my memories… it was a wasteland, and people used to throw rubbish in there… and now it’s a beautiful oasis in Liverpool 8!’(P3, M, 63, Group 1) (Figure 4)

Another interesting aspect that emerged from this study is the role that blue spaces played in participants’ perception of social inclusion (in this case, the River Mersey). Walking along the river not only created a sense of attachment to the aesthetic aspect of the river (Figure 5), but it was perceived as an integral part of participants’ sense of identity towards Liverpool.
‘I think that’s a wonderful idea and a real achievement for Liverpool. It gets me out in the fresh air… I like to walk anyway, but how pleasant is to walk there along the river? […] this is about the feel-good factor that’s so important in life.’(P20, F, 81, Group 4) (Figure 5)
‘The river has always been a part of my life. It just feels Liverpool when you stand by the river […]’.(P12, M, 68, Group 2)

The combination of accessibility, affordability and sociability of green and blue spaces contributed to participants’ mental wellbeing, feeling of inclusion and sense of connection.

#### 3.2.2. Transportation

Alongside green and blue spaces, public transportation was perceived as a key resource that strengthened participants’ sense of inclusion and respect. It was necessary for older people’s ability to remain independent, and to participate regularly in community life. Overall, participants felt very satisfied with the public transport system in Liverpool. It was perceived as efficient and accessible, giving them the freedom to reach local places as well as more iconic places in the city centre (Figure 6).
‘I really appreciate our very good bus and train services. […] I can come out in the morning and I can get on a bus and go to so many places, and I don’t have to wait long for a bus. It gives me freedom: freedom to get out. We are so lucky! Buses and trains are such a vital part in our lives.’(P20, F, 81, Group 4) (Figure 6)

The affordability of transportation options was essential, particularly for those less willing to drive, with no access to a car, or on a low income.
‘The free travel pass helps me to connect with the community that I live in. […] it allows me to travel across Merseyside for free and do things that I would not otherwise be able to do […]’(P3, M, 63, Group 1)

The free travel pass for people aged 60+ covers buses, trains and ferries across the areas surrounding Liverpool (called Merseyside), offering additional opportunities to access places that are not necessarily in the local ‘community’.
‘In Liverpool, we can use the buses but [also] the trains and the Mersey Ferry, […] … we’ve got a lot more on offer with our bus pass than people in other parts of the country.’(P17, F, 64, Group 3)

Accessibility was not only perceived in terms of the transportation itself, but also *how* participants accessed it. The quotes and photos below show two contrasting examples of local stations, and how they enabled or hindered participants’ sense of respect and social inclusion. The first one provided a safe and warm environment for people waiting for the bus or train (Figure 7). Moreover, the integration of different public transports offered more options and made it easier to travel.
‘I am very lucky because I have got all of that near to me, and it gives me access to a lot of options to travel. We’ve got the trains in and out and the buses in and out… it just makes it so much easier for people to travel. Once you get inside the building, you’re protected from the weather and the wind, it is very cosy, convenient, and very accessible. Older people deserve the respect and a good travel experience.’(P1, 65, M, Group 1) (Figure 7)

By contrast, the second bus station (located at the heart of the city) was perceived as very uncomfortable due to lack of protection from the wind, and not particularly safe (Figure 8).
‘Growing older is not about changing a lot…or being in a transition… but it’s about keeping the opportunities the same for you as they are for everybody else. So, things like this [transportation] become very important. As you can see, Liverpool One bus station is completely open, and the wind can still get in! We should use the learning from South Parkway station [Figure 7], and apply it to this station, so that you’re behind closed doors, in comfort, while waiting for a bus.’(P1, 65, M, Group 1) (Figure 8)

Both photos were taken by the same participant, who used photographs and their associated meanings to illustrate the concept of accessibility as an important dimension of social inclusion in the urban context.

As shown in the quotes related to Figure 6 and Figure 7 (and already in Figure 3), participants described themselves to ‘be lucky’. This seems to indicate a positive outlook for these participants and a strong personal attachment to the city.

#### 3.2.3. Additional Public Facilities: Public Toilets and Pavements

Participants identified other public facilities that shaped their perceptions of accessibility. These included public toilets and pavements. Shortage of toilets, especially in spaces regularly used by the community, was reported as preventing many people from going out, impacting negatively on their ability to feel confident in public spaces (Figure 9).
‘This toilet’s not inviting, it’s not accessible…and I’ll be a bit anxious about getting locked in… It’s counterproductive to meeting the needs of older people. Your body is changing and you’ve different needs…, but you need immediate access to clean toilets. It’s against social inclusion. It’s a barrier because not many older people will be confident to go to the city centre if there’re not enough accessible public toilets.’(P1, 65, M, Group 1) (Figure 9)

The same participant who took the photo in Figure 9. Suggested that premises in the city centre (e.g., cafes and restaurants) could allow older people provided with the free travel pass to use their toilets:
‘[…] the city council could have a scheme whereby if you are an older person with a bus pass, or if you have disabilities and you have your car badge, they should negotiate with all the restaurants and cafes, and pubs that people won’t be stopped from accessing those toilets.’(P1, 65, M, Group 1)

Tactile paving at the end of the sidewalks was identified as particularly important not only for older adults, but for everyone with lower levels of mobility and/or disability. Being able to walk safely in the neighbourhood was perceived as having a positive impact on social participation, as people could get out more often (Figure 10).
‘These little bumps ensure safety when there’s bad weather. For some people, just stepping off the side is not much, but when you’ve got bad legs, it is awful; so, if you have these bumps to help you getting around, then it helps you in the community; you can actually get out more, and you can socialise a bit more […]’.(P23, F, 70, Group 4) (Figure 10)

#### 3.2.4. Disrespectful Environmental Attitudes and Sense of Disregard and Alienation towards the Community

Whilst many aspects of the urban space helped participants feel valued and part of their community, poorly maintained environments (uncleanliness, litter on the streets and general decline of the neighbourhood) contributed to feelings of vulnerability and a sense of exclusion. It is interesting to note that all the photographs and quotes relating to negative perceptions of the physical environment originated from participants living in more disadvantaged areas (Table 3). This may suggest that participants living in more affluent areas did not identify or perceive similar issues in their neighbourhoods. In fact, they reported positive perspectives of their neighbourhoods.

According to some participants, the uncleanliness of the streets was mainly to do with lack of respectful attitudes of some people towards their local community and failure of the Council to address this. The litter in the street was not only seen to lower the tone of the neighbourhood. It was perceived as a bad example for young people, meaning they did not learn how to be respectful towards their community and, by implication, its residents, including older adults (Figure 11).
‘This lowers the tone of the neighbourhood. I think I live in a quite nice neighbourhood […] the bin man’s just been…but that mess is still there. [It] does not engender any respect in the young. The young see that and think “well, everyone else is throwing away their rubbish”. […] The people who live there should be aware of it […] There’s no respect for yourself or for anyone else with that around.’(P20, F, 81, Group 4) (Figure 11)

The following quotes show a close interlink between aspects of the physical environment (e.g., litter on the streets) and the social environment (e.g., a lack of respect for the older generation) that contributed to feelings of vulnerability and a sense of exclusion. At times, participants’ efforts to address disrespectful attitudes led to a more direct experience of disrespect, as shown in this quote:
‘The other day a young woman was eating a package of crisps. She finished the crisps and threw the package on the floor. Five yards away there was a rubbish bin. I said to her: “why you don’t put the package on the rubbish bin?” And she told me to f**k off. So, I said ok, bye! It’s a very disappointing aspect, but… never mind.’(P3, M, 63, Group 1)

On other occasions, some participants reported that people living in more affluent areas often displayed stigmatising attitudes towards the more disadvantaged areas (where some study participants lived). Such episodes contributed to a perceived sense of exclusion and frustration, given that many felt unable to change the situation, for instance, by going to live in a nicer neighbourhood:
‘[…] I have been on the bus sometimes and I have been hearing people [from a particular affluent area] saying: “How do people live here? […] sometimes there are troubles and gangs”. I would like to hear people saying there’re good people here as well, and a lot of us have no choice to live anywhere else and move out.’(P21, F, 75, Group 4)

Living in a deprived neighbourhood not only adversely affected the attachment that participants felt towards these spaces, but in some instances, caused a feeling of alienation.
‘I feel alienated by the community when I see rubbish in the streets.’(P3, M, 63, Group 1)

Participants used their photos and narratives to raise awareness of some issues that they felt were not currently addressed by the Council and/or were the responsibility of residents.

### 3.3. Social Environment

As reported earlier on, participants identified a close connection between the physical and social dimensions of their living environment, with aspect of the physical environment (e.g., parks) contributing to many social aspects of their lives (e.g., meeting people).

#### 3.3.1. Places to Cultivate Learning, Art and Culture

Places to cultivate learning, art and culture were the second most photographed and talked about aspect by participants. With only one exception (Section 3.4), participants took photographs of libraries located in their area of residence, suggesting that libraries tend to be facilities mainly used by people who live locally. In a similar way to green spaces, libraries were free and accessible facilities to cultivate interests, as well as meeting places to open to everyone. Furthermore, this example shows how the aesthetic of the library contributed to the sense of attachment perceived towards Liverpool Figure 11.
‘This is the library. I like books and inside it’s absolutely beautiful… we have such a lovely facility here […] it’s lovely to have a look and see whatever you want to see… it’s open for anyone in Liverpool to go in, so it’s not local community but it’s for the community of Liverpool’.(P15, F, 64, Group 3) (Figure 12)

Alongside libraries, museums were identified as very valuable assets in the city. In these examples, a participant identified three main aspects that facilitated ease of access and frequency of use: (i) accessibility and affordability of museums (Figure 13) and (ii) the aspect of proximity to the local community (Figure 14).
‘This is Liverpool Museum… it’s just a lovely facility. All the museums that we have in Liverpool… they’re all free.’(P17, F, 64, Group 3) (Figure 13)
‘This is Sudley House. It’s been kept as a 19th century house; it has also very nice art inside. […] They also do Shakespeare’s plays in there in the summer. I use it because it’s a local facility for me, and I can walk there.’(P15, F, 64, Group 3) (Figure 14)

#### 3.3.2. Places to Cultivate Informal and Formal Relationships

When reflecting about spaces to cultivate informal and formal relationships, participants stressed the value of community centres. Community centres were often described as an inclusive space for social interaction, which contributed to a sense of wellbeing and feeling valued.
‘[…] It gives me something to live for, something to look forward to. It gets you out and it’s another reason to get up in the morning. It makes you feel good because we’re all nice people, and we all talk and have ideas together. We help each other.’(P22, F,67, Group 4) (Figure 15)

Soup kitchens, instead, were considered a good alternative for those who were not interested in joining a club:
‘I go there [to the Soup Kitchen] every week […] it makes people who wouldn’t normally come out to meet up with people. Lots of people don’t want to join a club… this is a nice way to have a bowl of soup, and some nice crusty bread, […] and you can just talk to the people of your table […]!’(P20, F, 81, Group 4)

In terms of public spaces, benches at the bus stops or in green spaces were key places for informal socialisation, where people could rest and have a chat with others.
‘This is the new bus shelter and [the council] provided seating again. […] that is very good because […] you can rest while you’re waiting for the bus, you can get chatting to people, and you also are covered from the wind. […] it’s part of the community because you get to be more sociable if you’re sitting down, and you will talk to people’.(P23, F, 70, Group 4) (Figure 16)

#### 3.3.3. Negative Age Perceptions and Disrespectful Attitudes towards Older Adults and Ageing as a Barrier

Despite that many aspects made participants feel valued and included, disrespectful attitudes towards older adults contributed to feelings of vulnerability in the community and a sense of exclusion.
‘More respect for older people. I think that’s where we are lacking now.’(P21, F, 75, Group 4)

Some participants reported instances that they felt intensified negative perceptions of ageing in society. The quote below shows how the language used to refer to older adults was perceived as disrespectful:
‘A lot of it it’s about feeling valued. I really don’t like being called an old, aged pensioner […] I like the phrase elder. In Australia, we call older people ‘elder’ because we respect them, and I really think that’s missing in English society’.(P3, M, 63, Group 1)

In fact, although older adults made useful contributions to society (e.g., voluntary work), it was perceived that Liverpool and society in general tended to consider people to have a value only in terms of working life (economic value):
‘Not just in Liverpool, within Western culture, we don’t value older people. It’s almost like that you retire, and you don’t work anymore for money, so you have no value in society, while we have a great deal of resources up here. […] we have a great deal of knowledge and wisdom that we could pass on young people if they want to listen to us.’(P3, M, 63, Group 1)

However, participants did challenge these negative perceptions through photographs and associated narratives. For instance, some facilities (e.g., allotments) were supporting people with lower mobility or forms of disabilities to carry out daily hobbies and activities in the community (Figure 17).
‘This [photograph] shows that people can still do outdoor activities and grow for themselves vegetables…and disability or immobility is not gonna stop them doing it! […] they are raised beds: if you’re old and you can’t get down, you can sit next to the beds and it makes it much easier […].’(P9, M, 71, Group 2) (Figure 17)

There were fewer photos portraying negative social aspects such as disrespectful attitudes towards older adults, although these were reported extensively during the SSIs and FGDs. Participants reported that they found it more difficult to take photographs of negative social concepts (e.g., social isolation) compared with negative physical aspects (e.g., rubbish in the street) or positive social concepts (e.g., social participation) (findings presented in our companion paper) [65].

#### 3.3.4. Neighbourhood Fragmentation and Lack of Social and Intergenerational Interactions as a Barrier

Wider social processes (e.g., family and neighbourhood fragmentation) were reported to lead to difficulty in cultivating intergenerational relationships, as families increasingly lived away from each other. Lack of knowledge exchange between generations was linked to negative age perceptions and disrespect, which participants believed could be addressed by encouraging contact between different age groups:
‘Young children can go and get to know older people, because there’s a lot of them that don’t talk to older people. If their own grandparents have died, they don’t get used to know older people...they just see old people as being old and miserable. But when they get to know them, they’d realise that they’re not. It’s like old people thinking young boys with hoods on are all bad children, but they’re not.’(P23, F, 70, Group 4)

Linked to this, participants highlighted that in the past people used to know their neighbours more than today, and they used to mix with each other. The sense of community was perceived as getting lost. People were constantly moving houses and out of communities, and this made it difficult to cultivate trusting relationships with neighbours and a community in which older people could feel embedded.
‘[…] what we lost is possibly the community aspect where people in the road looked after other people.’(P15, F, 64, Group 3)
‘[…] because families lived very close to each other, there was a very close community. I have just found out the name of the lady opposite to me! And I have been in that house for 17 years….’(P20, F, 81, Group 4)

This included reduced opportunities for older adults to informally meet young people in the neighbourhood. The problem was seen to be exacerbated by more people using cars for their transport and children now typically playing inside.
‘I used to walk to school, and you met people; but people come out of their house now, get into a car and go […].’(P11, F, 64, Group 2)
‘I have lived in the same street now for 36 years, and I knew everyone. I knew all the kids, but now the kids are all indoors, you don’t see them, so it’s a ghost town.’(P21, F, 75, Group 4)

This concern was not echoed by all participants. Some of those from a more affluent area stressed that there was a sense of strong community in their streets with good neighbourhood relations leading to a feeling of inclusion:
‘Our road is very neighbourly; people are around all the time. We know all our neighbours, we say hello to them, it may be not much more than that but […] we do see them, and our next door is actually one of our friends, […] so I think that depends on the road.’(P10, F, 67, Group 2)

### 3.4. Services

#### Communication and Access to Information

Access to the Internet (IT) and computing skills was an important aspect of participants’ perceptions of social and digital inclusion. IT was the most common means of accessing information by the study participants. IT was perceived as particularly important not only to access information more easily, but also to counteract some practical challenges such as not being able do shopping due to health issues:
‘When you have difficulties, as I did, I have done the online shopping and it was a great help to me. When I am fit enough to go out, I will go out shopping… but when you know how to use Internet, it makes you independent.’(P22, F, 67, Group 4)

IT training was often facilitated by community organisations (Figure 18) and libraries (Figure 19), which created a supportive environment in which older people felt encouraged to learn.
‘Going to this computer class with the other girls, they’re all the same age as me and I do not feel embarrassed. It’s very helpful, particularly shopping online. I was very ill last year, and I could not get out; people had to do my shopping, but it would have been easier if I could have done the shopping online.’(P21, F, 75, Group 4) (Figure 18)
‘This is our library, and anybody can go there and read the books… if you want to learn computers…they can teach you. It’s nice and quiet, and it’s a mixed community that goes in there. It’s a place to meet up with each other and discuss things, and you can make new friends in there!’(P2, F, 64, Group 1) (Figure 19)

However, a lack of computing or IT skills stood in the way to participation in the community, as access to information was limited.
‘It’s about knowing what’s going on even in the neighbourhood down the road or across the park… it’s still a problem if you have missed out on the technology thing.’(P1, M, 65, Group 1)
‘We live in a society which just uses computers all the time […] so if you haven’t got a computer, or if you aren’t computer literate in my age group, it’s a barrier’.(P17, F, 64, Group 3)

Despite IT being commonly used to access information, local leaflets and free newspapers held value for older adults in increasing their awareness of activities in their local area and city. To make access to information more inclusive for those who were not using IT, and to increase social inclusion through engagement with activities, participants suggested that organisations could put some leaflets and/or small display boards in the bus stops, post offices and supermarkets to advertise weekly events and activities. In fact, these amenities were accessed daily by most people.
‘They should advertise [events] more: in the papers, in the supermarkets, because […] everybody goes in the supermarkets […].’(P4, F, 70, Group 1)
‘[…] all the local shops could have just little weekly or monthly summary on what’s going on in the immediate neighbourhood in the next few weeks. So, if you go to the post office in the area, or you’re going to the local shop, or supermarket, you are going to be able to see it in the notice board.’(P1, M, 65, Group 1)

## 4. Discussion

To our knowledge, this is the first photovoice study to actively engage community members in exploring their perceptions of factors influencing their respect and social inclusion in a city seeking to become more age-friendly/an AFC. This study provides novel insights into how respect and social inclusion are shaped by aspects of the physical and social environment, and the role that accessibility, affordability and sociability of physical spaces had on older adults’ wellbeing, feelings of inclusion, sense of independence and connection. Participants also identified some aspects of the physical (e.g., litter on the streets) and social environment (e.g., age-stereotypes) that contributed to feelings of vulnerability and a sense of exclusion. Wider social processes (e.g., family and neighbourhood fragmentation) also contributed to difficulties in cultivating intergenerational relationships.

In line with what others [20,22] have reported, our findings suggest that rather than a discrete entity, respect and social inclusion is highly interconnected with the other AFC domains, including social participation, communication and information, outdoor spaces, transportation and housing [20,42]. As such, it should be considered a guiding principle for cities aspiring to become more age-friendly. Cities need to ensure that respect and social inclusion is reflected in city-based policies, including local strategies for older adults as well as services in the community [16].

For example, our findings showed that parks and iconic places (e.g., museums) in the city provided an accessible (and free) space for multi-generational interactions, and for which older people felt a strong sense of connection. The sense of identity and inclusion expressed by participants was often connected with the participant’s aesthetic experience, particularly concerning outdoor spaces and architecture. This result is consistent with previous research [24,45,46], which highlights an interaction with the different aspects of the environment (e.g., green spaces, iconic buildings) fostering a sense of community connection and identity. Moreover, our study clearly showed that older adults accept that they themselves have a responsibility for their own wellbeing and a big role to play in ensuring they remain included. For example, by going out, joining a club, meeting people, practicing physical activity, doing volunteering, or learning something new. However, to be truly ‘age-friendly’, a city must provide an environment where older adults feel both respected and included, and support them in overcoming physical and social barriers that can limit their involvement in their communities [34].

In our study, older adults saw IT as important in improving their access and sense of connection with the communities and wider social networks. The provision of free IT training classes equipped them with the skills needed to negotiate opportunities within their city (e.g., being digitally connected, doing online shopping,). Accordingly, IT can play a critical role in increasing awareness of existing activities and promoting social inclusion [69]. Connectivity beyond IT was still valued by participants and highlights the need for wider communication and knowledge sharing. Our results are consistent with other studies that have stressed the importance of sharing local information through the use of leaflets or notice boards displayed in places regularly used by older adults (e.g., supermarkets, bus stops and post offices) [22,70].

Accessibility and affordability emerged as key features of inclusive environments. Affordable transportation and accessible stations enabled older adults to use and access places in the city and remain independent. From our findings, the free travel pass in Liverpool was highly valued and provided an opportunity for *all* people aged 60+ to reach key destinations in the city. An AFC should include provision of affordable and accessible transportation options in related policy to ensure that older adults maintain participation in social life [27,70,71]. Responding to the different transport needs of the population can also have important benefits on equity, which is key to create inclusive environments [71]. A primary challenge for many older adults is, in fact, to maintain mobility, regardless of their physical limitations or disabilities associated with advancing age [15,44]. A city with paved, flat, accessible pathways, sidewalks and transportation options enables older adults to be mobile and to participate in the community, as our findings illustrate.

Our findings showed some differences in perceptions among participants living in affluent or more disadvantaged areas (Table 3). For instance, those living in more disadvantaged areas generally reported feeling disappointed with the appearance of, and people’s attitudes in, their local area. This included disappointment in how clean the community was kept and a general decline of the neighbourhood. These reports created a sense of exclusion and frustration, which was exacerbated when people living in more affluent areas displayed stigmatising attitudes towards areas where some of the study participants lived. These findings have important implications when considering people’s perceptions of age-friendly environments and in the development of AFCs. They highlight the importance of paying attention to the diversity not only between cities, but also within cities. This does include the social exclusion experienced by many older adults living in disadvantaged areas [16,34]. Conversely, in our study, all participants living in more affluent areas reported positive perceptions and feelings of inclusion towards their local area.

In addition to the feeling of social exclusion that older people who reside in disadvantaged and/ or stigmatised neighbourhoods might feel, such environments can have negative impacts on health and quality of life [72,73], even from an early age [74]. With advancing age, people are likely to become more emotionally attached to their local community and homes. Therefore, if older adults live in an environment they feel is supportive, they are more likely to experience a sense of belonging, and this can improve their wellbeing [24]. If, instead, they live in an environment perceived as neglected, and poorly maintained, they are more likely to experience a sense of exclusion and a negative influence on their wellbeing. This can also reduce their social participation within and outside the community [74,75].

Cities committed to developing AFCs need to consider policies that respond to the unequal contexts and disadvantages experienced by older adults [76]. Taking an equity lens in developing AFCs and addressing social, gender, ethnic and other forms of inequalities will not only benefit older adults but all age groups, especially those most at risk of experiencing social exclusion and disadvantage [77].

### 4.1. Implications for Research, Policy and Practice

Recognising respect and social inclusion as fundamental principles for age-friendliness has implications for both policy and practice. Research in the related field of dementia friendliness provides relevant insights [3]. Buswell et al. [78] highlight raising awareness (in this case, of dementia) and, thus, generating understanding in the community as the necessary starting point for creating an environment that enables people affected by dementia to live well. In the context of AFCs, awareness raising relates closely to the strategic objective of combating ageism in the *WHO’s Global strategy and action plan on ageing and health* [39]. This entails, for instance, tackling misconceptions and negative attitudes on ageing and drawing attention to the contributions made by older adults.

In our study, participants reflected to what extent the city made them feel respected as older adults. A perceived decline in respect accompanied by negative perceptions of ageing in society were negatively impacting how some participants viewed themselves and the activities they thought they could engage with. Participants reported that to challenge macro-societal barriers such as negative perceptions of ageing, change needed to happen beyond the local level (e.g., improving the terminology used to refer to older people in the media). This finding highlights that AFCs do not operate in a vacuum, but there needs to be a match between what they are working towards and efforts at the national level and across many sectors. At the macro level, what is critical to this is addressing ageism and tackling the discourses that promote the persistence of age-stereotypes and a negative view of ageing [38]. Awareness raising to foster respect and social inclusion can thus be advocated as a foundational activity for aspiring AFCs. In practice, this might take the form of intergenerational projects and spaces that promote a cohesive community and public information campaigns that promote the contribution of older people in a city, coupled with advertising of the city’s AFC initiative [70].

Another fundamental aspect for the successful development of an AFC is to ensure that respect and social inclusion are reflected in older adults’ experiences of their city/community. This can be achieved through an active involvement of older adults in decision-making processes, including working in partnership with community members and multiple stakeholders (local policy makers, public, private and third sector). The cities of Manchester (UK), Brussels (Belgium) and Quebec (Canada) represent some exemplars where collaborative partnerships were key to the successful implementation of AFC models [16,79]. For instance, Manchester (UK) has used a co-productive approach to engage older people as co-researchers in exploring features of age-friendliness in their city, and findings have provided evidence to assist in development of their AFC initiative.

However, significant challenges stand in the way of those aspiring to create equitable AFCs, most notably competing priorities in relation to allocation of resources within the local authorities [16]. In times of austerity with limited budgets, interventions needed to achieve an AFC may be given a low priority. Liverpool, together with several cities in the UK and globally, have experienced significant reductions in funding allocated to services for older adults [34,80].

Liverpool’s AFC history dates from 2012, when its Mayor signed a pledge committing the city to become an AFC, in accordance with the WHO AFC initiative. In 2014, Liverpool joined the WHO’s Global Network of Age-Friendly Cities and Communities. Despite the initial commitment, in a context of budget reductions, the AFC initiative in Liverpool got off to a slower start than many had anticipated. In 2019, Liverpool’s AFC initiative has regained momentum with the appointment of an AFC Lead and with the Deputy Mayor reaffirming the pledge to become an AFC [81]. Supported by a steering group with representation from older residents and from sectors across the city (transportation, art and culture, health), work to enhance Liverpool’s age-friendliness has progressed in different ways. The AFC coordinator has indicated that our study findings are still relevant for Liverpool, and city developments are very much in line with our results [80].

For instance, many study participants highlighted the need to foster intergenerational relationships in the city. Recently commissioned projects in Liverpool include intergenerational initiatives bringing together socially isolated older adults and allow them to teach their life skills to parents and children in their community [82]. At the Liverpool City Region level, there are plans to display AFC window stickers in premises [83]. Inspired by Nottingham City Council’s work [84] (UK), premises displaying the AFC window sticker aim to provide a free seat, make toilet facilities available and offer tea, coffee, or a glass of water to older residents. This initiative will address lack of accessible public toilets in Liverpool city centre: a barrier to respect and social inclusion identified by our study participants.

Our study has also highlighted the need to incorporate older residents’ views in AFC plans and promote partnership work across sectors. According to the AFC lead, one of the notable changes has been Liverpool City Council’s approach to work, which is now more focused on considering older residents’ needs and working closely with third sector organisations [80]. Recently, Liverpool City Council has been working with an independent organisation, Health Watch, to identify residents’ views on priorities that need addressing. Engagement is happening through a wide range of methods, including face to face and telephone interviews, and surveys [80]. A current challenge in Liverpool, is, however, finding other ways to routinely engage with a wider range of older residents, particularly those most at risk of experiencing social exclusion and disadvantage [80]. This highlights a common challenge for other cities developing AFC (or related) approaches: the need to find more effective ways of engaging with older adults. Our study has demonstrated the appropriateness of taking a CBPR approach in accessing older people to elucidate their views on respect and social inclusion in the city. We believe that the periodic use of photovoice methods should be considered by policy makers and public health practitioners as a tool to maintain engagement with older people in identifying priorities for action and ensuring that their views are included in decision-making processes for their city.

### 4.2. Strengths and Limitations

Photovoice, with the combination of FGDs, SSIs and photos, brought to the surface older people’s views about what was most important for their respect and social inclusion in an aspiring AFC, providing us with a unique angle to the issue, whilst engaging participants throughout the research [57,64].

A potential limitation of this study relates to the gender imbalance in the sample (males: 7; females: 19). The guiding principle for recruiting participants was to have a mix of included and less included participants, rather than focusing on gender differences. Moreover, we realised that (i) our strategy to recruit participants from grassroots organisations in order to include a mix of more or less socially included older adults and (ii) our requirement for participants to physically attend the sessions and being able to walk (even if for short distances only) to take photographs, meant that we did not reach some of the most excluded older people. However, our sample included a mix of affluent and less affluent participants from a range of locations in the city, as well as different age ranges and a mixture of ethnic and cultural groups. Finally, whilst our study was not designed to produce generalisable findings (e.g., to other contexts in the UK or elsewhere), some aspects identified by older adults in Liverpool are potentially applicable to other contexts (e.g., age-friendly non-urban communities). Our study therefore offers an opportunity to apply the same or similar research methods to explore perceptions of respect and social inclusion among older residents living in smaller and rural age-friendly communities.

## 5. Conclusions

This study has provided novel insights into how respect and social inclusion are shaped by aspects of the physical and social environment, and the role that accessibility, affordability and sociability of physical spaces contributed to older adults’ mental wellbeing, feelings of inclusion, sense of independence and connection.

This research has shown that respect and social inclusion is a key domain of an AFC, and one that cuts across most—if not all—other domains identified in the original WHO framework for AFCs. As such, it should be a guiding principle for cities and communities aspiring to become more age-friendly. AFCs need to ensure that respect and social inclusion are appropriately reflected in city-based policies, including local strategies for older adults as well as services in the community. Our study has also shown the importance of perspectives about ageing and deprivation held more widely across society. The AFC movement could make a collaborative effort nationally and internationally to help society as a whole to evolve in these regards.

Cities and communities developing AFC (or related) approaches need to find better ways of engaging with older adults, including those most at risk of experiencing social exclusion and disadvantage. The periodic use of photovoice methods should be considered by policy makers and public health practitioners as a tool to maintain engagement with older adults in identifying priorities for action and ensuring that their views are included in decision-making processes for their city.

## Figures and Tables

**Figure 1 ijerph-17-09246-f001:**
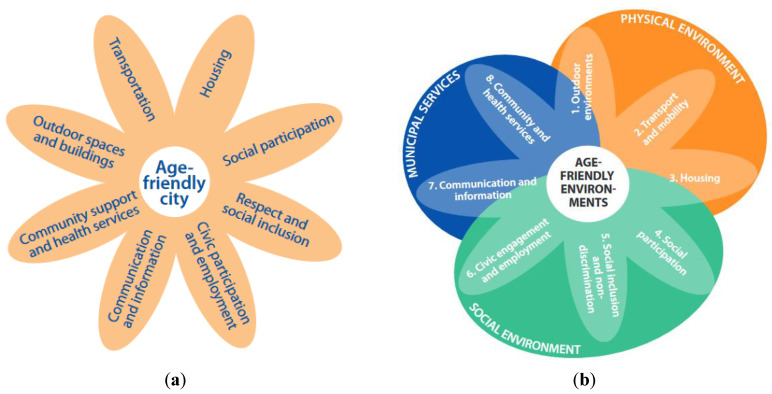
(**a**) Age-Friendly City (AFC) domains [10]. (**b**) Eight domains for age-friendly action [14].

**Figure 2 ijerph-17-09246-f002:**
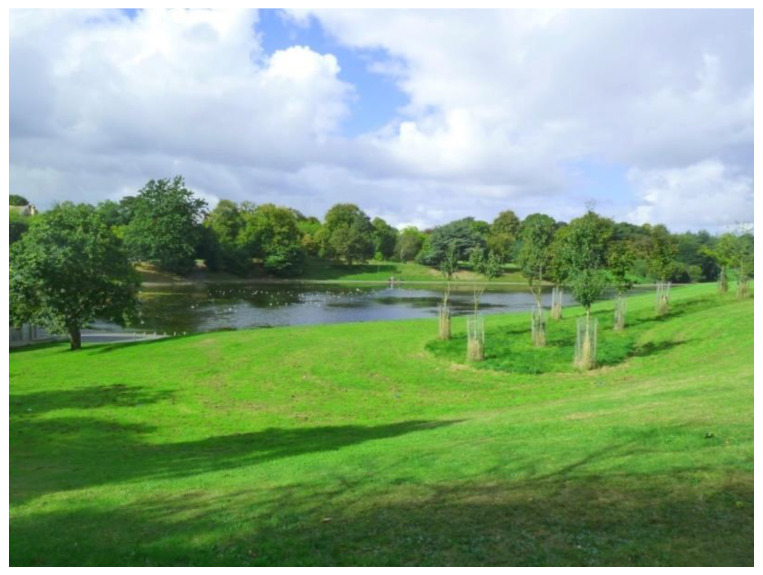
Example of green spaces promoting respect and social inclusion (P1) (Sefton Park, Liverpool).

**Figure 3 ijerph-17-09246-f003:**
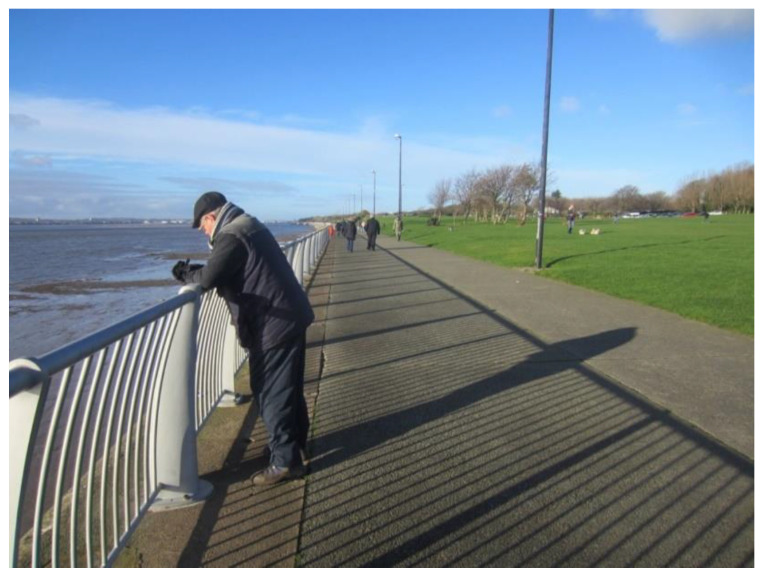
Example of green spaces promoting respect and social inclusion (P17) (Otterspool Promenade, Liverpool).

**Figure 4 ijerph-17-09246-f004:**
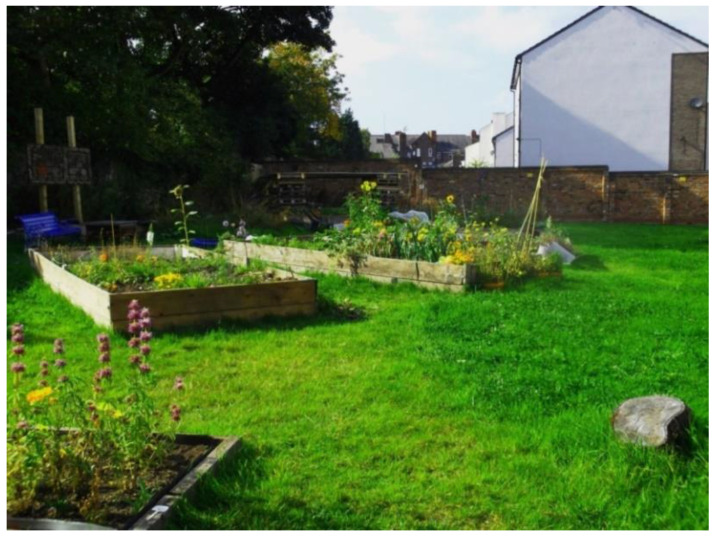
Example of a community garden promoting respect and social inclusion (P3) (Ferngrove community garden, Liverpool).

**Figure 5 ijerph-17-09246-f005:**
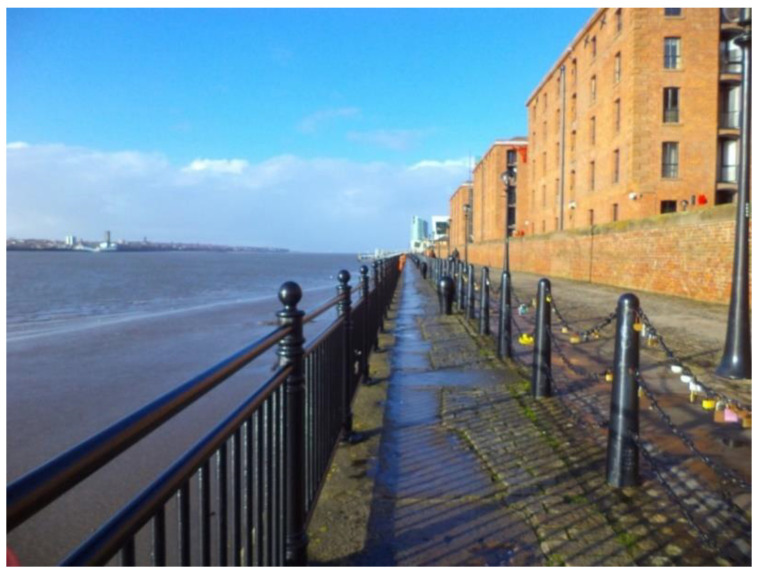
Example of blue spaces promoting respect and social inclusion (P20) (Albert Dock, Liverpool).

**Figure 6 ijerph-17-09246-f006:**
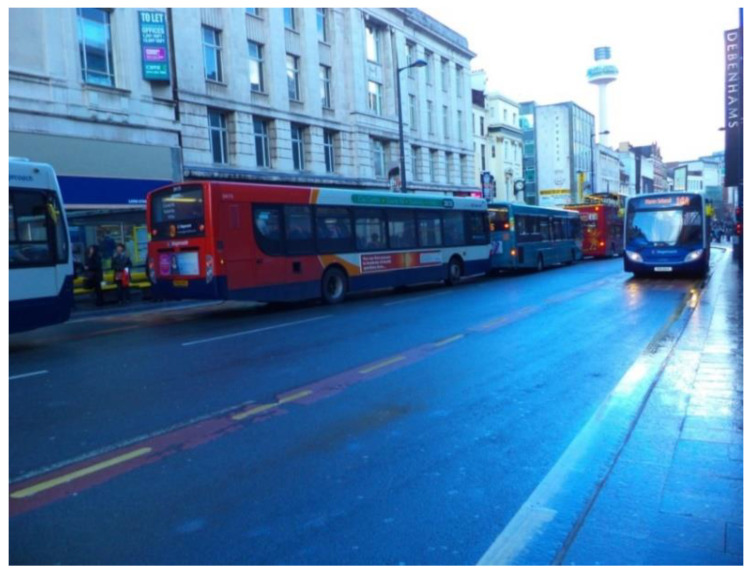
Example of transportation options promoting respect and social inclusion (P20) (Buses in Liverpool city centre).

**Figure 7 ijerph-17-09246-f007:**
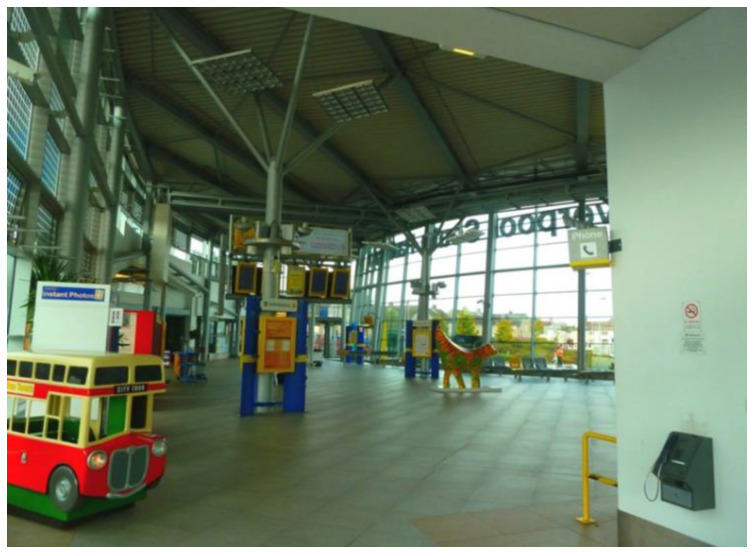
Example of train and bus stations promoting respect and social inclusion (P1) (South Parkway Railway Station, Liverpool).

**Figure 8 ijerph-17-09246-f008:**
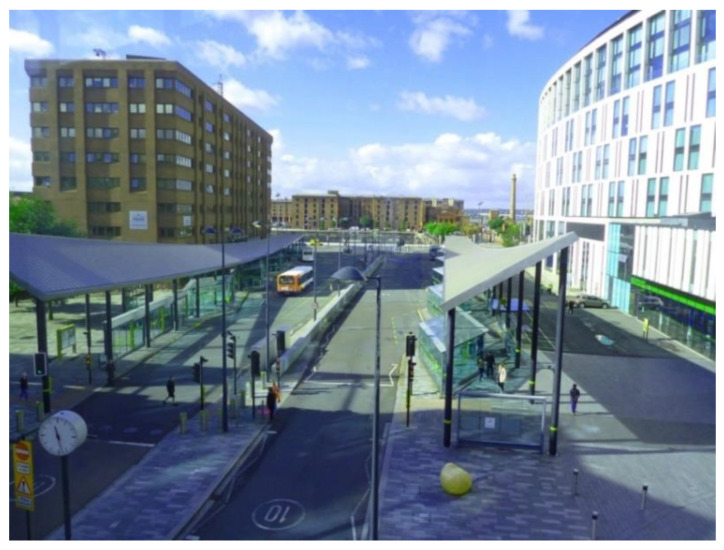
Example of bus stations hindering respect and social inclusion (P1) (Liverpool One bus station).

**Figure 9 ijerph-17-09246-f009:**
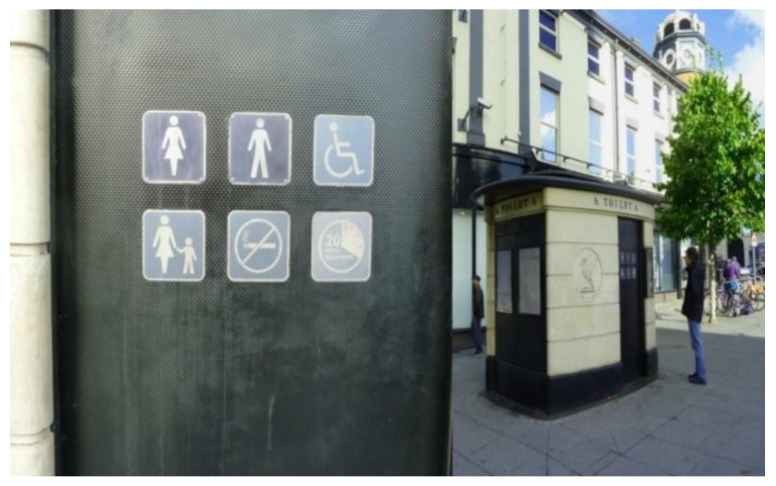
Example of public toilets hindering respect and social inclusion (P1) (Bold Street, Liverpool).

**Figure 10 ijerph-17-09246-f010:**
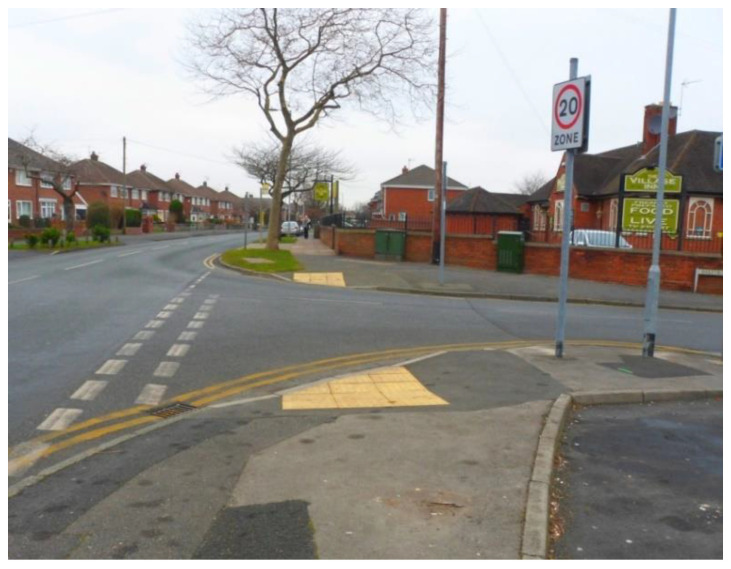
Example of sidewalks promoting respect and social inclusion (P23) (Road, Liverpool).

**Figure 11 ijerph-17-09246-f011:**
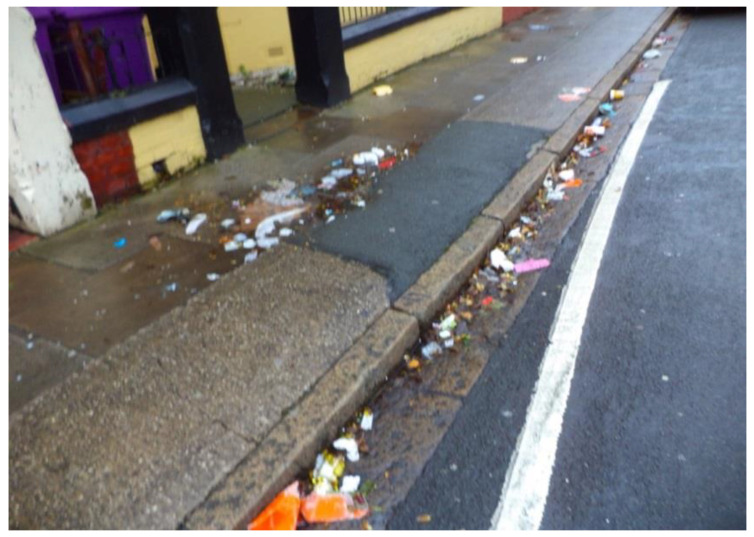
Example of litter in the street hindering respect and social inclusion (P20) (Road, Liverpool).

**Figure 12 ijerph-17-09246-f012:**
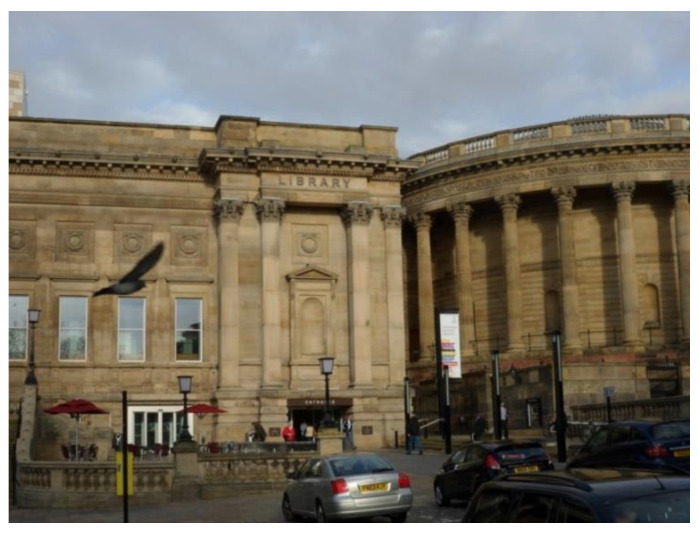
Example of libraries promoting respect and social inclusion (P15) (Central Library, Liverpool).

**Figure 13 ijerph-17-09246-f013:**
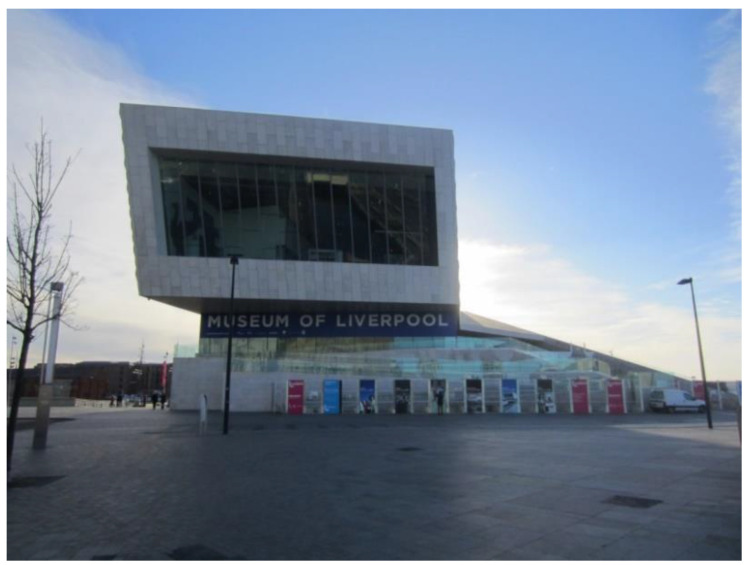
Example of museums promoting respect and social inclusion (P15) (Museum of Liverpool).

**Figure 14 ijerph-17-09246-f014:**
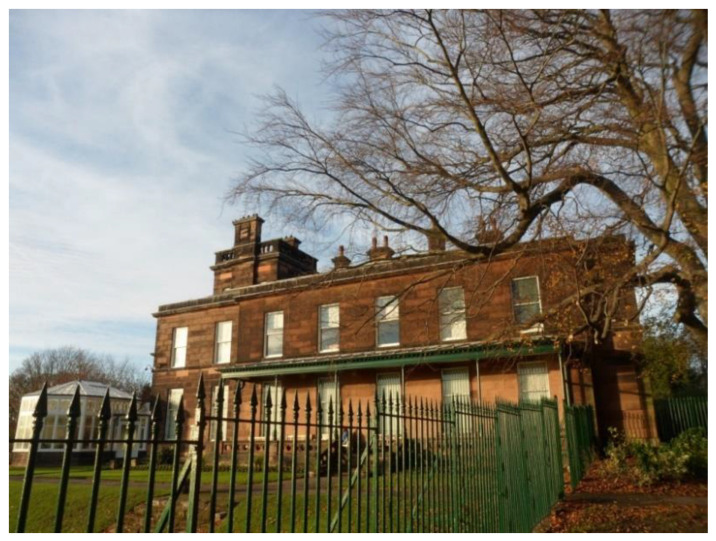
Example of museums promoting respect and social inclusion (P15) (Sudley House, Liverpool).

**Figure 15 ijerph-17-09246-f015:**
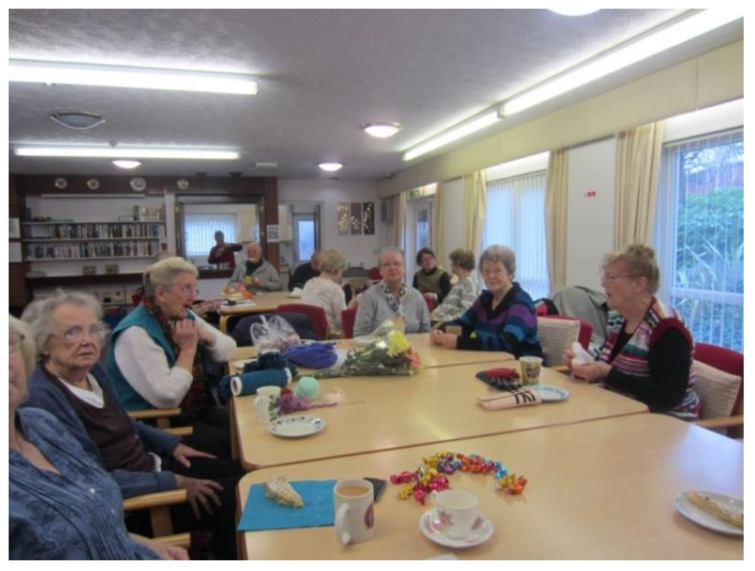
Example of a community group promoting respect and social inclusion (P22) (community group, Liverpool).

**Figure 16 ijerph-17-09246-f016:**
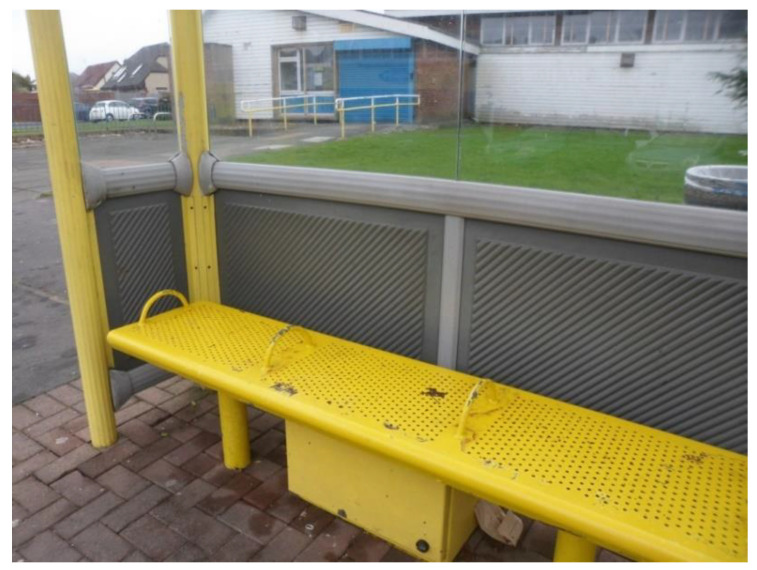
Example of a bus bench promoting respect and social inclusion (P23) (Bus stop, Liverpool).

**Figure 17 ijerph-17-09246-f017:**
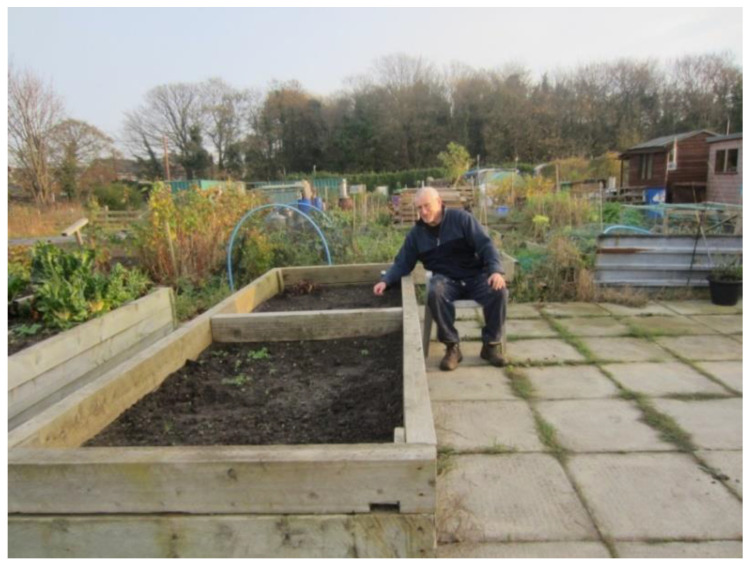
Example of allotments promoting respect and social inclusion (P9) (Allotment in Mersey Road, Liverpool).

**Figure 18 ijerph-17-09246-f018:**
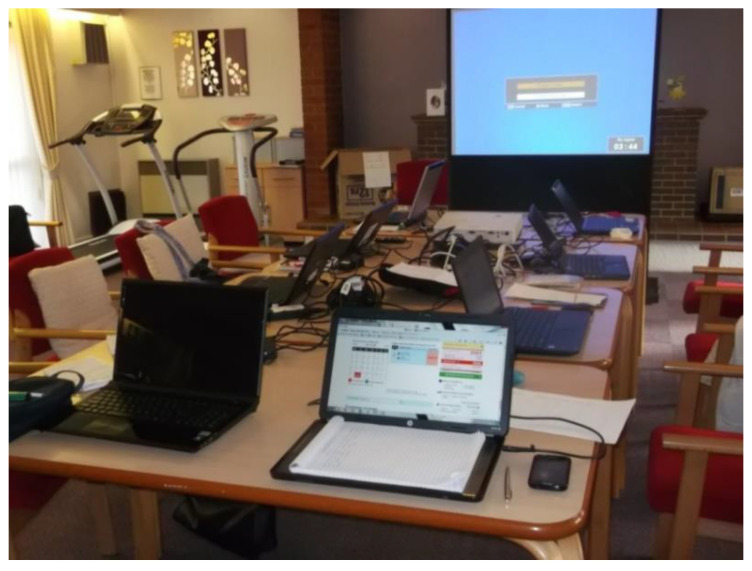
Example of a computing classes promoting respect and social inclusion (P21) (St. Luke’s court, Liverpool).

**Figure 19 ijerph-17-09246-f019:**
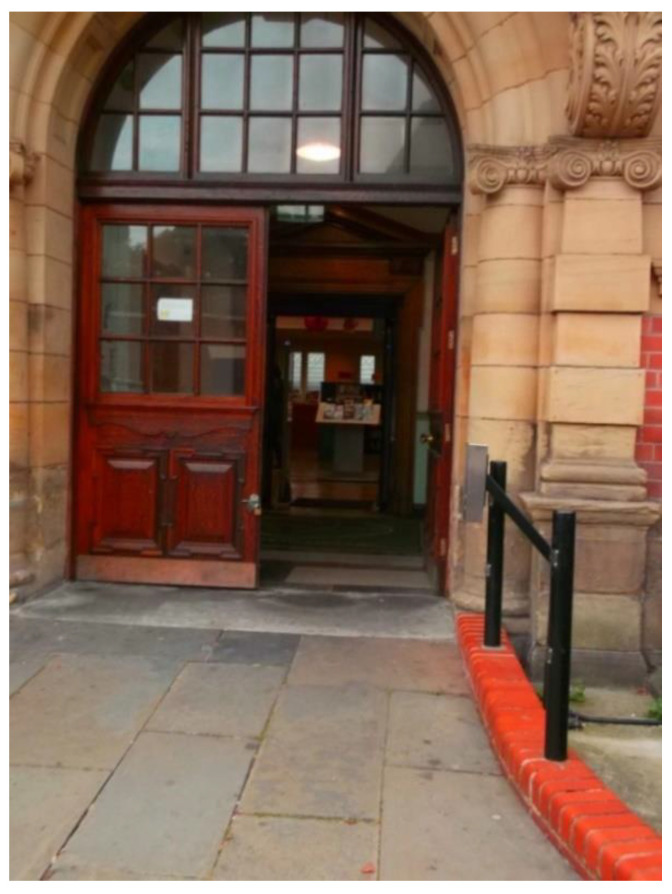
Example of libraries promoting respect and social inclusion (P2) (Toxteth Library, Liverpool).

**Table 1 ijerph-17-09246-t001:** Inclusion criteria of study participants.

1. Being able to consent for themselves.
2. Being an older person aged 60+ years.
3. Being able to speak English fluently.
4. Living in Liverpool.
5. Being British or having lived in the UK for at least 10 years.
6. Being able to manage simple digital cameras and take pictures about the topic under study.
7. Being able to attend and participate in group meetings and an individual interview.

**Table 2 ijerph-17-09246-t002:** Photovoice phases adopted in this study.

Photovoice Phase	Description
Phase 1: Photographic and ethical training session and initial Focus Group Discussion (FGD)	Overview of the project aims. Distribution of the digital cameras to participants with training. FGD explored general perceptions of respect and social inclusion in Liverpool. The FGD was audio recorded with permission. Participants were asked to photograph aspects of their environment that they felt ‘enabled or prevented feeling valued and part of the community’ and to identify potential solutions to any problems identified. Participants could photograph any object/person/place that referred to their views of respect and social inclusion in the city and neighbourhood. Participants did not receive examples of potential photos that they could take, in order to reduce the researcher’s influence over the participants’ choice of photographs [60,61,62]. Photographic and ethical training, including photo ownership and ethical implications of individuals appearing in photos [63,64]. Participants were given the ‘rules’ on when consent was needed (e.g., when a person or group is ‘featured’) and when not (where individuals can be considered a crowd). Participants were instructed to inform every person who appeared in the photographs on the study aim and that the photos were to be used as a part of publications and photo-exhibitions [62].
Phase 2: Taking the photographs	Participants took photographs over a period of approximatively a week.
Phase 3: Individual Semi-Structured Interview (SSI)	Each participant was shown their photographs on a laptop and asked to select approximatively six photographs they wanted to discuss at the interview and in the subsequent focus group. Restricting the number to six enabled a more in-depth exploration of each photograph [59]. Participants took part in an SSI (audio-recorded with permission). Questions explored the meanings associated with each photograph and informed by the *SHOWeD* technique [56]. The *SHOWeD* technique consists of different questions that relate to the photograph: ‘What do you See here? What’s really Happening here? How does this relate to Our lives? Why does this problem, concern, or strength Exist? What can we Do about it?’. Other questions were adapted from reviewing the literature on ageing and AFCs (Appendix A). To explore unrecorded issues of perceived importance, participants were asked to discuss any photograph that they had wanted to take, but for different reasons had been unable to [63,64].
Phase 4: Second FGD	Participants took part in a second FGD, where they collectively interpreted the photographs, including similarities and differences among images [65]. They identified key themes emerging from the discussion of the photographs. They then discussed how they wanted to communicate the findings to policy makers and relevant stakeholders (e.g., through a photo-exhibition event), including preferences for potential venues and plans for the photo-exhibition event. The FGD was audio-recorded with permission.
Phase 5: Summarising photos’ texts and checking these with participants	From transcripts, captions were developed by S.R. based on the participant’s explanation of each photograph. To guarantee that each caption accurately reflected the correct meanings, each participant reviewed and approved the captions in advance of the photo-exhibition [60]. Participants also reviewed and agreed on the selection of the photographs and accompanying texts to display in the photo-exhibition.
Phase 6: Disseminate the findings and advocacy (e.g., photographic exhibition)	In total, seventy-one people attended a public and stakeholder event, including participants, community members, representatives from Liverpool City Council, services for older people in the city, local TV and radio journalists, and academics. Twenty-three participants (out of 26) presented their photographs and narratives to the attendees of the event. Sixty-one photographs (out of 127) were displayed, with each participant having between two and three photographs and accompanying texts exhibited. The photographic exhibition provided a forum for participants to (i) directly communicate their views to city stakeholders and (ii) stimulate discussion of their priorities for healthy ageing and the way forward for respect and social inclusion in the city.

**Table 3 ijerph-17-09246-t003:** Study participants. M = male, F = female.

Group	Area	Level of Deprivation	N	Gender	Age Group	Ethnic Background
1	A	High	10	3 M, 7 F	60–64: 2	5 White British, 2 Asian British, 3 Black African British
					65–70: 2	
					70–75: 2	
					75–80: 1	
					80–85: 3	
2	B	Low	4	2 M, 2 F	60–64: 1	4 White British
					65–70: 2	
					70–75: 1	
3	C	Low	6	2 M, 4 F	60–64: 2	5 White British, 1 Other White Background
					75–80: 2	
					80–85: 1	
					>85: 1	
4	D	High	6	6 F	65–70: 1	6 White British
					70–75: 3	
					75–80: 1	
					80–85: 1	

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
