# Peer review of "How is Respect and Social Inclusion Conceptualised by Older Adults in an Aspiring Age-Friendly City? A Photovoice Study in the North-West of England"

_ijerph, 2020, doi:10.3390/ijerph17249246_

Round 1

Reviewer 1 Report

I wonder whether the term ageism should be explained, or maybe replaced. It sounds like political activism to me.

The present study seeks to investigate the impact of the urban surrounding on the way in which elderly members of the community perceive a sense of exclusison and disrespectful treatment in connection with the city´s “furniture”.  The participants of the study were free to document their impression of a certain area by means of taking pictures. There were discussions in focus groups as well as semi-structured inteviews. The study is qualitative in nature and focusses on the participants authentic experience of their surrounding city . The authors show that the sources of problems affecting the elderly are manifold including socioeconomic (eg affordability) and structural (accessibility). However, little is said about more physical barriers, eg health problems in the sample, which probably create additional dissatisfaction with the community. Moreover, the study investigates a subjective stance which is probably not free from perspectives relating to psychosocial distress. Thus the surrounding which someone is complaining about might play the role of the “whipping boy”, whereas the dissatisfaction stems from other sources (eg social or relational problems).

I would find some more explanation on the social aspects of the integration interesting, i.e. other findings.

Moreover, I was wondering about other groups such as the disabled (some of whom are 60+ of course). Are there comparable studies which could be mentioned? Two of the participants had some minor form of disability, but this seems not to be relevant for the study results. Since only English people were enrollable, the question arises as to how non-British view the city. Or are the British likely rpresentative of the  non-Brits in this respect? .

What is the situation like for the elderly in smaller communities – is it rather worse or better?    Could the results be generalized to inform other countries, as well?

Were there any exclusion criteria? I would assume that the satisfaction of the elderly with their urban surrounding is somewhat dependent on their own condition. Eg for some it may be very important to find eg medical facilities not too far away. Hence not only what is made of concrete seems to be relevant for the study question. Is the satisfaction influenced by sociodemographic factors (eg divorced/widow)?

Reviewer 2 Report

This manuscript presents the results of a qualitative study to determine perceptions of various aspects of urban environment living among older adults that affect their feelings of respect and social inclusion by their community.  The authors provide detailed characterization of these perspectives among the older adults who participated derived from participatory research using photographs taken by these elders, as well as individual interviews, focus groups, and stakeholders meetings. The manuscript provides some limitations as well as future research directions and some potential implications for community efforts to support older adults in their communities and establish not just Age-Friendly Cities but national-level efforts to include and respect their older adult members.

The strengths of the manuscript include the several methods employed in this qualitative study to determine the perceptions and feelings of the older adults in Liverpool. In addition, the inclusion of older adults from multiple areas of the city, particularly the more affluent to the poorer areas, is a strength and provides a sample that is more representative and not skewed toward only a single set of older adults. Another strength is the active inclusion and collaboration of older adults in the research and their individual as well as group perceptions regarding what makes them feel respected and socially included (or not) in their community as well as their suggestions for remedies to perceived problems, exclusions and negative attitudes. The focus on Respect and Social Inclusion seems to be unique or uncommon in considerations of Age-Friendly Cities domains.

There are some issues that might be addressed.

  • Recruitment: (a) Participants were recruited from community centres during weekly activities. It is also suggested (line 151) that the recruitment "was assisted by local community organizations." Does this mean community centres in all cases or were participants recruited in some other way or with help from other organizations? (b) How exactly were the individuals recruited? Were they simply asked individually to consent to participation or were there other aspects to the actual recruitment? (c) What about those older adults who do not utilize community centres or their weekly activities? Relatedly, you indicated that 4 participants had mobility/walking issues, but what about those who lack transportation or are not able to walk with the assistance of strollers/walking sticks? Issues of representativeness of the sample beyond just the issue you raised for sex/gender could also be noted or addressed.
  • Timing of Study: It is noted that the study was conducted from 2013-2016. Have there been any changes in the intervening time that would have occurred to influence the findings? Were any of the suggestions or ideas observed in the study or conveyed to the stakeholders implemented in that time period that could be reported?
  • Study Location: The study was conducted in Liverpool, UK. Certainly the issues identified by the older adults there exist in most cities, not only in England but elsewhere across the world (along with specific issues that might exist in those cities that differ from Liverpool). The limitations, at least regarding the specific findings, might be noted as potentially not generalizable to other specific cities in various locations. However, it could also be pointed out that the methodologies used here could actively engage the elders of other cities to arrive at the specific situation for older adults there. Additionally, the findings are obviously related to Age-Friendly Cities, but there are clearly issues that could be addressed for rural and smaller community elders as well that could be identified by similar research methods (for "Age-Friendly Communities" perhaps).
